

# *EvatCrop*: a novel hybrid quasi-fuzzy artificial neural network (ANN) model for estimation of reference evapotranspiration

Gouravmoy Banerjee[1], Uditendu Sarkar[2], Sanway Sarkar[3] and Indrajit Ghosh[1]

[1] Department of Computer Science, Ananda Chandra College, Jalpaiguri, West Bengal, India
[2] National Informatics Centre, Ministry of Electronics & Information Technology, Government of India, Kolkata, West Bengal, India
[3] Ernst & Young LLP, Bengaluru, Karnataka, India

## ABSTRACT

Reference evapotranspiration ($ET_0$) is a significant parameter for efficient irrigation scheduling and groundwater conservation. Different machine learning models have been designed for $ET_0$ estimation for specific combinations of available meteorological parameters. However, no single model has been suggested so far that can handle diverse combinations of available meteorological parameters for the estimation of $ET_0$. This article suggests a novel architecture of an improved hybrid quasi-fuzzy artificial neural network (ANN) model (*EvatCrop*) for this purpose. *EvatCrop* yielded superior results when compared with the other three popular models, decision trees, artificial neural networks, and adaptive neuro-fuzzy inference systems, irrespective of study locations and the combinations of input parameters. For real-field case studies, it was applied in the groundwater-stressed area of the Terai agro-climatic region of North Bengal, India, and trained and tested with the daily meteorological data available from the National Centres for Environmental Prediction from 2000 to 2014. The precision of the model was compared with the standard Penman-Monteith model (FAO56PM). Empirical results depicted that the model performances remarkably varied under different data-limited situations. When the complete set of input parameters was available, *EvatCrop* resulted in the best values of coefficient of determination ($R^2$ = 0.988), degree of agreement ($d$ = 0.997), root mean square error ($RMSE$ = 0.183), and root mean square relative error ($RMSRE$ = 0.034).

Corresponding author
Indrajit Ghosh, ighosh@accollege.in

## INTRODUCTION

The agriculture sector of India consumes a considerable amount of groundwater resources for irrigation (*Khajuria, Yoshikawa & Kanae, 2013*; *Dhawan, 2017*). The replacement of groundwater mostly depends on rainfall. Unfortunately, a negative trend in the annual rainfall is observed in India (*Radhakrishnan et al., 2017*), which is detrimental to the groundwater reserve. Moreover, the increasing demand for food crops and irrigation water
creates additional stress on groundwater resources (*Roy, Nilson & Pal, 2008*). In the Terai agro-climatic region of North Bengal in India, a massive amount of groundwater is used for irrigation, especially during the rabi season (November to May). Due to the lack of knowledge for precise estimation of irrigation water, most layman farmers use traditional flood irrigation methods and waste a considerable amount of precious groundwater, which creates a major problem of moderate to extreme groundwater stress during the rabi season (*Sahoo et al., 2021*). Therefore, a model for precise estimation of crop water requirement for efficient irrigation practices is now very significant for conserving groundwater resources and sustaining agriculture.

Reference evapotranspiration ($ET_0$) is a significant parameter for measuring the crop water requirement. Several methods and models have been suggested worldwide for the estimation of $ET_0$ using region-specific meteorological parameters. Among them, the Food and Agriculture Organization recommended the Penman-Monteith method (FAO56PM) as the benchmark method for estimating $ET_0$ (*Pandey, Dabral & Pandey, 2016*), and its performance was validated against lysimetric data in diverse climatic conditions worldwide (*Xu et al., 2013*; *Ghamarnia et al., 2015*). However, the major limitation of the FAO56PM method is that a complete set of meteorological parameters that are required for estimating $ET_0$ are challenging to obtain or may not be available for most regions in developing countries (*Djaman, Irmak & Futakuchi, 2017*; *Wu et al., 2020*). To overcome the data inadequacy problem, several empirical models based on limited meteorological parameters were suggested, such as the Hargreaves-Samani model (*Hargreaves & Samani, 1985*), the Priestly-Taylor model (*Priestley & Taylor, 1972*), the Irmak model (*Irmak et al., 2003*), the Makkink model (*Makkink, 1957*), *etc*. However, due to region-specific limitations and poor portability, these models are not good enough for the efficient estimation of $ET_0$ (*Wu et al., 2020*). Therefore, alternative and efficient methods that estimate $ET_0$ within a tolerable range using limited meteorological parameters are needed (*Mattar, 2018*; *Kisi & Alizamir, 2018*).

As an alternative, several artificial intelligence (AI) techniques were proposed for estimating $ET_0$ (*Heddam et al., 2018*). These techniques were more popular than direct and indirect methods due to their capacity to tolerate incomplete or imprecise information. At an early stage, as a popular AI technique, several fuzzy inference systems (FIS) were attempted for $ET_0$ modeling, those included (*Odhiambo, Yoder & Yoder, 2001*; *Patel, Patel & Bhatt, 2014*; *Faybishenko, 2012*), *etc*. However, these models did not survive due to severe drawbacks, such as domain experts' scarcity, rule ambiguity, and the improper design of the membership functions (*Wang, 2015*). Several variants of decision tree (DT) based algorithms were attempted to design $ET_0$ estimation models with fewer inputs. *Kisi (2016)* suggested three different regression approaches to estimate $ET_0$. *Salahudin et al. (2023)* used a decision tree (DT) model along with a random forest and a boosted tree model to predict $ET_0$ using limited climatic inputs for Pakistan. The other contributions were made by *Feng et al. (2017)*, *Fan et al. (2018)*, and *Huang et al. (2019)*. However, there are some general problems with these decision tree-based models. They require a large amount of data (*Ehteram et al., 2019*), a large tree structure for proper modeling of the $ET_0$ (*Üneş, Kaya & Mamak, 2020*), and suffer from overfitting (*Elbeltagi et al., 2022*).

Moreover, they cannot express complex relations between the input and the output. Many investigators have used artificial neural network (ANN) to model $ET_0$, such as *Bruton, McClendon & Hoogenboom (2000)*, *Kumar et al. (2002)*, *Xu et al. (2006)*, *Chattopadhyay, Jain & Chattopadhyay (2009)*, *Yassin, Alazba & Mattar (2016)*, *Antonopoulos & Antonopoulos (2017)*, *Üneş et al. (2018)*, *Gocić & Arab Amiri (2021)*, *Kaya et al. (2021)*, *Pinos (2022)*, *Azzam et al. (2022)*, *Dadrasajirlou et al. (2022)*, *Heramb et al. (2023)*, and *Patle et al. (2023)*. Researchers reported that, like a decision tree, the ANN-based models are also prone to overfitting and require large datasets for their training (*Pinos, 2022*). Furthermore, there is no standard method to determine the structure of a neural network for modeling a problem (*Azzam et al., 2022*). Instead of using a standalone application, several hybrid systems endeavoured to provide better solutions. Among them, neuro-fuzzy systems have gained popularity in various complex domains. A typical fuzzy system is hybridised with an ANN to design a neuro-fuzzy system that combines the benefits of fuzzy logic and artificial neural networks. It is evident that, compared to a standalone fuzzy system or an ANN, hybrid neuro-fuzzy systems, especially adaptive neuro-fuzzy inference systems (ANFIS), confirmed better accuracy for $ET_0$ modeling (*Ladlani et al., 2014*; *Kisi et al., 2015*; *Patil & Deka, 2017*; *Salih et al., 2019*; *Kaya & Taşar, 2019*; *Üneş, Kaya & Mamak, 2020*; *Güzel et al., 2023*). However, the ANFIS models suffer from several significant drawbacks. They are data-driven models without a physical basis (*Tabari et al., 2013*). The entire learning process of ANFIS is fixed; it is impossible to manipulate the mapping of the input to the output (*Üneş, Kaya & Mamak, 2020*). Moreover, there is no established rule for modeling the structure of an ANFIS model (*Aghelpour, Bahrami-Pichaghchi & Kisi, 2020*).

Although the accuracies of the existing models were good enough, no study has reported an alternative architecture and application of a hybrid model for a more precise estimation of $ET_0$. In this article, we propose a novel architecture of a hybrid model for efficient estimation of daily evapotranspiration, targeted for crop irrigation (*EvatCrop*). Our proposed model was applied for estimating $ET_0$ in the Terai agro-climatic region of North Bengal, India, and compared with the other three popular models, DT, ANN, and ANFIS, using the same datasets. The empirical results demonstrate that *EvatCrop* excelled over DT, ANN, and ANFIS for modeling daily $ET_0$.

The specific objectives of this study were: (1) to suggest a novel architecture of a neuro-fuzzy model for better estimation of $ET_0$, (2) to study the model performance under various input combinations of meteorological parameters for $ET_0$ estimation in the Terai agro-climatic region of North Bengal in India, and (3) to compare the performance of our proposed model with other three prevailing models, DT, ANN, and ANFIS.

## MATERIALS AND METHODS

### Study area

For the case study, the proposed model was applied in the Terai agro-climatic region of North Bengal, India. This area (9,614 sq. km) is located in the northern part of the state of West Bengal in India, from 25.94 °N to 27.00 °N and 88.41 °E to 89.87 °E (*Mandal et al., 2022*). Figure S1 (*ESRI, 2009*), presents a geographic map of the study area. This region is

characterized by humid to hot sub-humid weather conditions. The minimum and maximum monthly average temperature during the rabi season (November to April) is about 13.9 °C and 27.2 °C, respectively, and during the kharif season (May to October), it is about 23.2 °C and 29.7 °C, respectively. The average monthly rainfall in this region is around 37.6 mm during the rabi season and 543 mm during the kharif season. The average monthly wind speed during the rabi season is between 1.0 and 1.2 m/s, whereas during the kharif season, it is between 0.88 and 1.97 m/s. The northern part is covered with forests, and the rest of the area (6,055 sq. km) contains agricultural lands with sandy clay loam to silty clay loam soil textures. The livelihood of the people of this area is primarily based on agriculture, resulting in intensive agricultural activities. Most cultivations in this area depend on groundwater irrigation during the rabi season. Rural farmers use traditional flood irrigation methods without precisely estimating the actual crop water requirement. Such indiscriminate use of groundwater resources threatens groundwater conservation. A recent study reported that a large part of this region is now subjected to moderate to extreme groundwater stress during the rabi season (*Sahoo et al., 2021*). Based on the availability of authenticated datasets, we selected three study locations dispersed across the Terai agro-climatic region of North Bengal, India. The three study locations are Berubari (latitude 26.38 °N, longitude 88.75 °E), Jayanti (latitude 26.70 °N, longitude 89.69 °E), and Tamaguri (latitude 26.07 °N, longitude 89.38 °E), as shown in Fig. S1 (*ESRI, 2009*).

## Datasets

The study used five meteorological parameters as inputs to estimate $ET_0$ (mm/day). The five inputs were the minimum atmospheric temperature ($T_{min}$) (°C), maximum atmospheric temperature ($T_{max}$) (°C), wind speed ($W_s$) (m/s; at 2 m above ground), relative humidity ($R_h$) (%), and solar radiation ($S_r$) (MJ m$^{-2}$ d$^{-1}$). The latest available meteorological datasets for three locations were obtained from National Centres for Environmental Prediction (NCEP) Climate Forecast System Reanalysis (CFSR) databases that provide benchmarked datasets (https://swat.tamu.edu/data/cfsr). The time variant data of each input parameter and that of the reference $ET_0$ of the three study locations are presented in Figs. S2 to S4.

No preprocessing was carried out as there were no missing values or outliers observed in the datasets. The statistical measures, such as minimum (Min), maximum (Max), standard deviation (Sd), coefficient of variation (Cv), and skewness coefficient (Sk), of each input parameter are presented in Table S1. It was observed that the skewness coefficient of wind speed ($W_s$) has a positive value, while other parameters have both positive and negative skewness coefficients. The lowest coefficient of variation was recorded for the maximum temperature.

The acquired datasets were divided into two parts: 70% for training (from 1st January 2000 to 16th March 2010), consisting of 3,278 samples, and the rest 30% for model testing (from 17th March 2010 to 31st July 2014), having 1,598 samples. The training sets were used for model construction, while the test sets were used to evaluate the model's performance.

## MATERIALS AND METHODS

### Combinations of input parameters

Due to various constraints, the complete dataset of all input parameters may not always be available from all study locations. In such circumstances, limited input parameters are available instead of a complete set of five input parameters. Our model was designed to deal with various combinations of five meteorological parameters: $T_{min}$, $T_{max}$, $W_s$, $R_h$, and $S_r$. A total of eight possible combinations of five input parameters were considered to estimate $ET_0$ for three study locations. The eight possible input combinations ($C1$ to $C8$) of meteorological parameters are presented in Table S2. The two input parameters $T_{min}$ and $T_{max}$ are measured using a common instrument (a thermometer) and are always available in pairs. This pair of input parameters was considered the base combination.

### Reference values of $ET_0$

*EvatCrop* was designed based on the supervised learning algorithm. For proper training and testing of the model, the estimated values of $ET_0$ were compared with the reference (target) values of $ET_0$. The FAOPM56 method is recognized as the only standard method to compare the performances of any other models used for estimating $ET_0$ (*Quinn, Parker & Rushton, 2018*; *Valipour, Gholami Sefidkouhi & Raeini-Sarjaz, 2020*; *Sarma & Bharadwaj, 2020*). The reference values of $ET_0$ (mm/day) were calculated using the FAOPM56 method (*Allen et al., 1998*):

$$ET_{0PM} = \frac{0.408 \times \Delta \times (R_a - G) + \gamma \times \frac{900}{T+273} \times u_2 \times (e_s - e_a)}{\Delta + \gamma \times (1 + 0.34 \times u_2)} \tag{1}$$

where the variables $R_a$, $G$, $T$, $u_2$, $e_s$, $e_a$, $\Delta$, and $\gamma$ bear the same meaning and units as previously described in *Banerjee, Sarkar & Ghosh (2022)*. All these variables were estimated using 24 additional equations involving different atmospheric inputs such as temperature, humidity, wind speed, *etc.*, as suggested by *Allen et al. (1998)* to estimate $ET_0$.

### Model architecture

Unlike a typical neuro-fuzzy system, our proposed *EvatCrop* model is a neuro-quasi-fuzzy system. The novelty in the architecture of this model is that an artificial neural network (ANN) is hybridized with a quasi-fuzzy inference system instead of a classical fuzzy inference system. It consists of two modules: a quasi-fuzzy module and an ANN module. The quasi-fuzzy module is coupled with an ANN module in such a way that it accepts the meteorological parameters as inputs and processes them to give the interim outputs. These interim outputs are then fed to the ANN module as inputs. After processing, the ANN module provides the estimated value of $ET_0$ as the final output. This model utilizes the merits of both fuzzy systems and neural networks. The model architecture and the process flow diagram are presented in Figs. S5 and S6, respectively. The model was implemented using the Scikit-learn library package in Python.

## Quasi fuzzy module

A typical fuzzy inference system (FIS) consists of four functional components: a fuzzifier, a knowledgebase, an inference engine, and a defuzzifier (*Odhiambo, Yoder & Yoder, 2001*). The fuzzifier transforms the real numerical inputs into relevant fuzzy sets. The knowledge base contains the rule base (a group of control rules) and a database that defines the information about the membership of each fuzzy set. The inference engine fires some selected rules based on the control strategy and generates fuzzy outputs. Finally, the fuzzy output is transformed into numerical output by the defuzzyfier. Our proposed quasi-fuzzy module does not have any defuzzifier component. The defuzzifier component is replaced by a normalization component that normalizes the outputs produced by the inference engine before feeding them to the ANN module. An input interface unit is designed to accept any input combination out of the eight combinations ($C1$ to $C8$).

### *Fuzzyfication*

The membership functions of each of these five input variables were constructed from the datasets of each input parameter. Three fuzzy linguistic values or labels; (*low*, *medium*, and *high*) were considered for each of these five input parameters. Fuzzification was achieved by characterizing input data space for each parameter into three fuzzy labels; *low*, *medium*, and *high*. The degree of membership ($\mu$) of data points in the respective fuzzy labels was determined by three membership functions as presented by Eqs. (2)–(4).

$$\mu_{low}(x) = \begin{cases} 1, & x = l \\ \frac{m-x}{m-l}, & l < x < m \\ 0, & x = m \end{cases} \tag{2}$$

$$\mu_{medium}(x) = \begin{cases} 0, & x = l \\ \frac{x-l}{m-l}, & l < x < m \\ \frac{h-x}{h-l}, & m < x < h \\ 0, & x >= h \end{cases} \tag{3}$$

$$\mu_{high}(x) = \begin{cases} 0, & x <= m \\ \frac{x-m}{h-m}, & m < x < h \\ 1, & x >= h \end{cases} \tag{4}$$

where $l$ and $h$ are the minimum and maximum values of each parameter observed in the respective data space and $m = (h - l)/2$. The shape of the membership functions of each parameter, their degree of overlap, and data spaces were elicited through the review of literature, interviews with experts, and current agricultural practices. The fuzzy membership functions of five input parameters for three study locations are presented in Figs. S7–S9.

### *Fuzzy rule base and inferencing*

The rule base consists of rules with an antecedent-consequent or IF-THEN structure. The antecedent (IF part) is formed by all possible combinations of premises obtained from

five input variables with three fuzzy labels (*low*, *medium*, and *high*). The premises are combined using a fuzzy union operator (AND). The consequent part is obtained by applying fuzzy product triangular norms (t-norms) operations (*Chien, 1990*; *Bag & Samanta, 2015*). The product t-norm is one of the popular methods used for merging the values of two or more fuzzy sets into a single value and can efficiently combine the criteria in multi-criteria decision-making (*Hájek, 1998*). It is applied in control to formulate assumptions of rules as conjunctions of fuzzy premises.

For example, the structure of the $i-th$ rule $(R_i)$ for three input parameters $T_{min}$, $T_{max}$ and $W_s$ is defined as:

$R_i \rightarrow$ IF ($T_{min}$ IS *low*) AND ($T_{max}$ IS *medium*) AND ($W_s$ IS *medium*) THEN $P_i$.

where $P_i$ is the product t-norm defined as:

$$P_i = \mu_{low}(T_{min}) \times \mu_{medium}(T_{max}) \times \mu_{medium}(W_s) \tag{5}$$

For eight possible combinations of input parameters, a total of 575 rules were framed. These 575 rules were divided into eight rule groups ($R_{G1}$ to $R_{G8}$). Each rule group contained all possible unique rules designed for the respective combination of input parameters. Some examples of rules are:

Rule 16: $\rightarrow$ IF ($T_{min}$ IS *low*) AND ($T_{max}$ IS *medium*) AND ($W_s$ IS *high*) AND ($R_h$ IS *low*) THEN $P_{16} = \mu_{low}(T_{min}) \times \mu_{medium}(T_{max}) \times \mu_{high}(W_s) \times \mu_{low}(R_h)$

Rule 49: $\rightarrow$ IF ($T_{min}$ IS *medium*) AND ($T_{max}$ IS *high*) AND ($W_s$ IS *medium*) AND ($R_h$ IS *low*) THEN $P_{49} = \mu_{medium}(T_{min}) \times \mu_{high}(T_{max}) \times \mu_{medium}(W_s) \times \mu_{low}(R_h)$

Rule 65: $\rightarrow$ IF ($T_{min}$ IS *high*) AND ($T_{max}$ IS *medium*) AND ($W_s$ IS *low*) AND ($R_h$ IS *medium*) THEN $P_{65} = \mu_{high}(T_{min}) \times \mu_{medium}(T_{max}) \times \mu_{low}(W_s) \times \mu_{medium}(R_h)$

In a typical fuzzy system, some selected rules are activated to give the output. Unlike a typical fuzzy system, in our proposed quasi-fuzzy module, all the rules in a rule group are activated to generate a set of outputs. As the rule base contains the complete set of all possible rules, the dependency on the judgment of any human expert for rule optimization is circumvented. A rule group selector was incorporated to select the respective rule group to be activated for a particular input combination.

## Normalization unit

The quasi-fuzzy module is designed by replacing the defuzzifier of a typical FIS with a normalization unit. As the rules and their numbers are different in each rule group, the proportionate contribution of each rule should be considered. The proportional contribution is obtained by normalization. The normalization unit calculates the proportionate contribution of the output of each rule in a particular rule group. This normalized output of a rule is considered an interim output. For the $i-th$ rule with output $P_i$, the normalized output is:

$$I_i = \frac{P_i}{\sum_{i=1}^{N} P_i} \tag{6}$$

where $N$ is the total number of rules in a rule group ($R_G$).

After normalization, the set of rules in a rule group produces a set of normalized outputs. For convenience, the set of normalized outputs is called the interim output vector $\{I\}$. The interim output vector $\{I\}$ is considered as input to the ANN module. The rule groups along with the number of rules in each rule group for each set of input parameters are presented in Table S3.

*Artificial neural network module*

Multilayer perceptrons (MLP) are one of the most popularly used ANN models (*Almeida, 2020*). In general, MLP models are trained using supervised learning algorithms such as backpropagation, Levenberg-Marquardt, L-BFGS, stochastic gradient descent, adaptive moment estimation, *etc.* The MLPs have a wide range of applications in hydrological research, including streamflow forecasting (*De Faria et al., 2022*), rainfall forecasting (*Diop et al., 2020*), monthly pan evaporation prediction (*Zounemat-Kermani et al., 2021*), *etc.* The ANN module in our proposed system consists of eight MLPs, each having an input layer, a hidden layer, and an output layer of nodes (*Cinar, 2020*). Each MLP is designed to accept the interim output vector $\{I\}$ produced by a particular rule group ($R_G$).

For optimization, each MLP was trained using stochastic gradient descent (SGD), Adam, and the limited-memory Broyden-Fletcher-Goldfarb-Shanno (L-BFGS) optimization algorithms (*Byrd et al., 1995*). Empirically, it was observed that the MLPs trained with the L-BFGS algorithm led to more accurate results as compared to the others. The L-BFGS optimizer belongs to the quasi-newton family of optimizers, with significant advantages over the two other training algorithms, as it provides a stable solution and requires less hyperparameter tuning. In L-BFGS optimization, the training begins by initializing the weights with small random numbers. After each iteration, the weights are updated using Eqs. (7) and (8) (*Byrd et al., 1995*).

$$w_{k+1} = w_k + \eta_k d_k \tag{7}$$

$$d_k = -H_k \nabla E(w_k) \tag{8}$$

where $w_k$ is the weights at iteration $k$, $\eta_k$ is the learning rate, $d_k$ is the direction of search, $H_k$ is the inverse of the estimated Hessian matrix, and $\nabla E(w_k)$ is the partial derivative of the error function. The best value of other hyperparameters of the MLPs *i.e.*, the activation function and the number of neurons in the hidden layer were obtained by a grided search method. The optimum values are provided in Table S4.

## Performance metrics

Evaluation of the performance of a model and a comparative analysis with other contemporary models play a significant role in any model-building process. For evaluation and comparison of the performances of different models, several metrics have been suggested in the literature. Instead of a single metric, a combination of multiple metrics is more trustworthy for evaluating a model (*Chai & Draxler, 2014*). In this article, a well-accepted performance evaluation strategy was adopted where the performance of each model was evaluated in terms of four common metrics; coefficient of determination ($R^2$), degree of agreement ($d$), root mean squared error (*RMSE*), and root mean squared relative

error (*RMSRE*). $R^2$ and *d* are two goodness metrics that measure the model's accuracy. The $R^2$ projects the relation between the actual and estimated values, and *d* measures the consistency between them. *RMSE* and *RMSRE* are two error metrics that project the errors in the estimated values. The mathematical equations of the evaluation metrics are defined below (*Tao et al., 2018*; *Yaghoubi, Bannayan & Asadi, 2020*).

$$R^2 = \frac{\sum_{i=1}^{n} \left( ET_{0PM_i} - ET_{0_{P_i}} \right)^2}{\sum_{i=1}^{n} \left( ET_{0PM_i} - \overline{ET}_{0_P} \right)^2} \tag{9}$$

$$d = 1 - \frac{\sum_{i=1}^{n} \left( ET_{0_{P_i}} - ET_{0PM_i} \right)^2}{\sum_{i=1}^{n} \left( \left| ET_{0_{P_i}} - \overline{ET_{0PM}} \right| + \left| ET_{0PM_i} - \overline{ET_{0PM}} \right| \right)^2} \tag{10}$$

$$RMSE = \sqrt{\frac{1}{n} \sum_{i=1}^{n} \left( ET_{0PM_i} - ET_{0_{P_i}} \right)^2} \tag{11}$$

$$RMSRE = \sqrt{\frac{1}{n} \sum_{i=1}^{n} \left( \frac{ET_{0PM_i} - ET_{0_{P_i}}}{ET_{0PM_i}} \right)^2} \tag{12}$$

where $ET_{0PM_i}$ is the reference value of $ET_0$ obtained using the FAOPM56 method, $ET_{0_{P_i}}$ is the estimated value, $\overline{ET_{0PM}}$ is the mean value of reference evapotranspiration obtained using FAOPM56 method, $\overline{ET_{0_P}}$ is the mean predicted value of reference evapotranspiration and *n* is the total number of input patterns.

For a comprehensive study of the performance of our proposed model in comparison to other prevailing models, two other metrics, average goodness (*Ag*) and average error (*Ae*) were used. Where *Ag* is the average value of two goodness metrics: coefficient of determination ($R^2$), and degree of agreement (*d*), and is defined as:

$$Ag = \frac{R^2 + d}{2} \tag{13}$$

Similarly, *Ae* is the average value of two error metrics: root mean squared error (*RMSE*) and root mean squared relative error (*RMSRE*), and is defined as:

$$Ae = \frac{RMSE + RMSRE}{2} \tag{14}$$

A higher value of *Ag* indicates a better goodness of fit, while a lower value of *Ae* indicates a better performance of a model.

## RESULTS ANALYSIS

In order to compare *EvatCrop* with other models for estimating daily $ET_0$, we developed three other prevalent machine learning models: a decision tree (DT), an artificial neural network (ANN), and an adaptive neuro-fuzzy inference system (ANFIS). All the models were compared together using the same training and testing datasets obtained from three study locations, Berubari, Jayanti, and Tamaguri, in the Terai agro-climatic region of North Bengal. The standard reference values of $ET_0$ ($ET_{0PM}$) were calculated using the

FAO56PM method (Eq. (1)). The tools and techniques used to develop the DT, ANN, and ANFIS models are described as an overview.

### Decision tree (DT)

A DT model uses a greedy technique to build a decision tree from root to leaf nodes. A feature that best divides the data into two or more parts is used as the root node of the tree. Thereafter, until a stopping criterion is met, the tree is developed by recursively splitting the data into smaller subsets depending on the feature values. The aim of a decision tree is to find the feature that yields the greatest information gain for each split, resulting in a tree that illustrates the relationships between characteristics and the target variable (*Duda, Hart & Stork, 2006*; *Murty & Devi, 2015*). For the present study, the DT model was developed using the scikit-learn library package in Python. The two hyperparameters: the maximum depth and the minimal cost-complexity pruning were adjusted using a fivefold grided search strategy using the training dataset. The range of maximum depth was set between 2 and 16 and the minimal cost-complexity pruning was set between 0.0 and 1.0.

### Artificial neural network (ANN)

Artificial neural network (ANN) models try to mimic the information processing techniques of a biological neuron (*Fausett, 2006*). A classical ANN model consists of processing units called artificial neurons, which are interconnected using synaptic weights, and it learns patterns from data by adjusting these weights. Various architectures were proposed to design an ANN using a variety of activation functions. For the comparative study, the ANN model was implemented using the scikit-learn library package in Python. A two-layer ANN model with one hidden layer and one output layer was developed. The activation functions, the values of hyperparameters, and the number of hidden layer neurons were adjusted using a five-fold grided search strategy using the training dataset. The outperforming activation functions were selected by trial and error. The number of hidden layer neurons was set between 2 and 16. Three learning algorithms: the adaptive moment estimation, the L-BFGS, and the stochastic gradient descent were used to train the model, and the best one was selected for the present study.

### Adaptive neuro fuzzy inference system (ANFIS)

The ANFIS model was first proposed by *Jang (1993)*. This hybrid model combines the benefits of fuzzy logic and artificial neural networks using a three-stage architecture. In the first stage, the input variables are transformed into fuzzy sets using membership functions. In the second stage, a decision is achieved by applying a set of IF-THEN rules to the fuzzy inputs. The final stage reduces the outputs of the previous stage to a single crisp value by using defuzzification. The parameters for the rule antecedents and membership functions are weighted to reduce the difference between what happened and what was predicted. Due to the interpretability of fuzzy logic and the capacity for learning and adaptation of an ANN, the ANFIS excels at solving complicated, non-linear issues that traditional rule-base systems find challenging (*Du & Swamy, 2013*; *Patel & Gianchandani, 2011*). For the present study, the ANFIS model was developed using the ANFIS toolbox

(https://in.mathworks.com/help/fuzzy/anfis.html) of MATLAB (version R2022a). The total number of fuzzy membership functions was set to three, and the parameters associated with membership functions (pre-parameters) of the developed ANFIS model were adjusted by a backpropagation algorithm on the training datasets.

### Estimating $ET_0$ of the Berubari location

The values of the four primary metrics ($R^2$, $d$, $RMSE$, and $RMSRE$) along with two combination metrics; average goodness ($Ag$) and average error ($Ae$) obtained for the four models (DT, ANN, ANFIS, and *EvatCrop*) at the training and testing phases for the Berubari location are summarised in Tables S5 and S6, respectively. All four models were compared in terms of $Ag$ and $Ae$ against different input combinations (from $C1$ to $C8$). The results presented in Table S5, depict that in the training phase, the *EvatCrop* was the most accurate model for all input combinations. The ANN was the worst model, whatever the input combinations. Table S5 depicts that for combinations $C2$, $C5$, and $C8$, ANFIS has the same accuracy compared to *EvatCrop* in the training phase. For other input combinations, DT and ANFIS fought neck-to-neck to secure second place. The results in Table S5 show that DT, ANN, and ANFIS were not quite as good as *EvatCrop*. They were about the same for input combinations $C2$, $C5$, $C6$, and $C8$ when it came to $Ag$ (differences ranging from 0.10% to 1.83%) and $Ae$ (differences ranging from 2.66% to 68.69%). The performances of all the models are very poor and unacceptable for $C1$, $C3$, $C4$, and $C7$. In conclusion, in the training phase, for five input combinations, the performances of DT, ANFIS, and the *EvatCrop* are comparable to each other. For the four input combinations $C6$, $C7$, and a three-input combination $C2$, the accuracy differences between all four models are also marginal, while the ANN is the worst model in all the cases.

In the testing phase, all the models were compared against different input combinations. Table S6 shows that *EvatCrop* did better than all the other models in terms of having the highest $Ag$ (from 0.675 for $C1$ to 0.992 for $C8$) and the lowest $Ae$ (from 0.116 for $C8$ to 0.737 for $C1$), no matter what combinations of inputs were used. It was the best performer with all input parameters (combination $C8$). However, the performances of other models vary with input combinations. In Table S6, we can see that DT, ANN, and ANFIS are not as good as *EvatCrop* when it comes to $Ag$ (differences range from 0.50% to 1.65%) and $Ae$ (differences range from 20.53% to 40.23%) for input combinations $C2$, $C5$, and $C6$. The DT model was the least accurate model for $C1$ (with $Ag$ = 0.650 and $Ae$ = 0.764), $C2$ (with $Ag$ = 0.971 and $Ae$ = 0.228), $C3$ (with $Ag$ = 0.672 and $Ae$ = 0.737), $C5$ (with $Ag$ = 0.967 and = 0.242), and $C6$ (with $Ag$ = 0.977 and $Ae$ = 0.201). For $C4$, ANN is the worst performing (with $Ag$ = 0.710 and $Ae$ = 0.705). Whereas, ANFIS was the least accurate model for $C7$ (with $Ag$ = 0.639 and $Ae$ = 0.814) and $C8$ (with $Ag$ = 0.658 and $Ae$ = 0.821). A possible reason is that the ANFIS model is developed using the grid partition approach, where the number of rules increases exponentially with the increasing number of input parameters, leading to increased computation time and reduced performance. A similar observation was reported by *Yeom & Kwak (2018)*. Additionally, the ANFIS model suffers from an out-of-range problem in the test set that further degrades its performance, as described by *Fu et al. (2020)*.

To study the impact of the meteorological inputs on model accuracy at the Berubari location, we provide a detailed analysis of the results during the testing phase using *EvatCrop* as the best optimum model. As reported in Table S5, using only two inputs, $T_{min}$ and $T_{max}$ (*C1*), the *EvatCrop* model provided the lowest accuracy (*Ag* = 0.675 and *Ae* = 0.737). Using three input variables, when wind speed $W_s$ was added with $T_{min}$ and $T_{max}$ (*C2*), the performances of the *EvatCrop* model increased significantly. The value of *Ag* was increased by about 31.00% from 0.675 to 0.981, and the value of *Ae* was decreased by almost 75.00% from 0.737 to 0.184. When relative humidity ($R_h$) or solar radiation ($S_r$) was added with two inputs ($T_{min}$ and $T_{max}$) instead of $W_s$ (*C3* and *C4*), the estimation accuracy of the *EvatCrop* model dropped significantly. The *Ag* value was cut down by 26.60% for combination *C3* and 36.18% for combination *C4*. Correspondingly, the *Ae* values were increased by 73.00% for combination *C3* and 72.00% for combination *C4*. When four parameters were used as inputs (combinations *C5*, *C6*, and *C7*), two combinations with $W_s$ (*C5* and *C6*) produced reasonably good results. The average value of *Ag* for *EvatCrop* against two combinations, *C5* and *C6*, was obtained as 0.988, which is almost 27.00 and 23.00% better than that of combinations *C3* and *C4*, respectively. Correspondingly, the average value of *Ae* for *EvatCrop* against two combinations *C5* and *C6* was observed to be 0.147, which represents a huge drop of 78.40% and 76.10% for combinations *C5* and *C6*, respectively. However, when $W_s$ was not present in the four-input combination (*C7*), the accuracy of all the models dropped significantly. For combination *C7* instead of *C5*, the value of *Ag* was cut down by 19.35% for *EvatCrop*. Similarly, when combination *C7* was taken instead of *C6*, the values of *Ag* were decreased by 20.00% for *EvatCrop*.

The results presented in Tables S5 and S6 indicate that better accuracy was achieved when the wind speed ($W_s$) was available as one of the input parameters. This observation specifies that the meteorological parameter wind speed ($W_s$) significantly contributes to estimating the $ET_0$ across the targeted study region. Similar observations were reported by *Traore, Wang & Kerh (2010)* in the Sudano-Sahelian zone of Burkina Faso, *Yang et al. (2019)* in China, and the literature survey conducted by *Amani & Shafizadeh-Moghadam (2023)*. Although the complete set of five input parameters provided the highest estimation accuracy for $ET_0$, fairly good performance was achieved with the other three input combinations, *C2*, *C5*, and *C6*. Based on estimation accuracy, the input combinations are ranked as *C8*, *C6*, *C5*, *C2*, *C7*, *C4*, *C3*, and *C1*. The results presented in Tables S5 and S6 summarise that (i) the *EvatCrop* was the most accurate model for estimating $ET_0$ at the Berubari location; (ii) the ANFIS model performed worst in the testing phase for combinations *C7* and *C8* due to the out-of-range problem; and (iii) the combinations that include $W_s$ as one of the input parameters (combinations *C2*, *C5*, *C6*, and *C8*) produce more accurate results than those without $W_s$. Figure S10 presents the scatter plots of reference values $ET_{0PM}$ *versus* the values of $ET_0$ estimated by the *EvatCrop* against eight input combinations. This figure shows that the highest performance for $ET_0$ estimation was achieved when all the input parameters were available. Figures S11 and S12 display the boxplot and Tailor diagrams, respectively of the four models for a better comparative

analysis of the results obtained in the testing phase. Additionally, the distribution plots for the estimated and reference $ET_0$ are provided in Fig. S13.

## Estimating $ET_0$ of the Jayanti location

Results obtained in the training phase at the Jayanti location are reported in Table S7. From Table S7, we observe that in the training phase, the *EvatCrop* model comparatively outperformed all other models for input combinations $C1$, $C3$, $C4$, $C5$, and $C7$, and for the other combinations, the DT model was the best. In the case of combinations $C2$, $C6$, and $C8$, DT performed 0.20% (2.84%), 0.30% (14.43%), and 0.50% (62.03%) better than the *EvatCrop* in terms of $Ag$ ($Ae$), respectively. It is to be noted that, in the training phase, the ANFIS model exhibited the same performance as *EvatCrop* in terms of $Ag$ for combinations $C5$, $C6$, and $C8$. For the other five combinations ($C1$, $C2$, $C3$, $C4$, and $C7$), the ANFIS and DT models secured the second position with nearly equal performances. For the combinations $C2$, $C5$, $C6$, and $C8$, the accuracy of all the models was similar in terms of $Ag$ (difference ranging from 0.10% to 2.10%) and $Ae$ (difference ranging from 3.53% to 80.64%). In conclusion, for combination $C8$, the DT, ANFIS, and *EvatCrop* models produced their best results, and their performances were comparable to each other. Whereas, the ANN model performed worst.

Table S8 summarizes the results obtained in the testing phase for the Jayanti location. The results indicate that *EvatCrop* performed best with the highest (lowest) average values of $Ag$ ($Ae$), irrespective of the input combinations. The accuracies of the other three models, DT, ANN, and ANFIS, are also comparable with *EvatCrop* for input combinations $C1$, $C4$, $C5$, and $C6$ in terms of $Ag$ (differences ranging from 0.20% to 2.05%) and $Ae$ (differences ranging from 0.79% to 53.28%). Except for input combinations $C7$ and $C8$, the DT model exhibited the poorest performance ($Ag$ ranging from 0.731 to 0.956 and $Ae$ ranging from 0.210 to 0.519).

To reveal the impact of the meteorological inputs on model accuracy, an in-depth analysis was performed based on the results summarised in Table S8 in the testing phase. The *EvatCrop* exhibited the highest level of accuracy at the Jayanti location when all five inputs ($T_{min}$, $T_{max}$, $W_s$, $R_h$, and $S_r$) were considered as inputs (combination $C8$). The input combination $C1$ with only two input variables ($T_{min}$ and $T_{max}$) produced the worst results ($Ag$ = 0.784 and $Ae$ = 0.504). From Table S8, it is observed that the differences in accuracy in terms of $Ag$ ($Ae$) in comparing $C1$ to $C8$ are 31.28% (73.41%). The value of $Ag$ ($Ae$) decreased (increased) by 12.83% (65.01%) when we removed $W_s$ from combination $C8$ (combination $C7$). But when we removed $R_h$ instead of $W_s$ (combination $C6$), the performance of *EvatCrop* marginally tarnished in terms of $Ag$ (decreased by 0.10%) and $Ae$ (increased by 2.19%). By removing $S_r$ from $C8$ (combination $C5$), $Ag$ was reduced by 3.76%, and $Ae$ was increased by 42.73%. It is to be noted that among the three four-input combinations, $C6$ produced the comparatively best accuracy, as observed in Table S8. For three input combinations, when we removed $W_s$ from combination $C6$ (combination $C4$), the performance of *EvatCrop* decreased further by 17.53% in terms of $Ag$ and increased by 62.78% with respect to $Ae$. The drop in $Ag$ and rise in $Ae$ were reasonably small when we replaced $S_r$ by $R_h$ in $C4$ (combination $C3$). The $Ag$ decreased by 2.71% and
the $Ae$ increased by 7.36%. However, when $R_h$ was replaced by $W_s$ in combination $C3$ (combination $C2$), the overall performance of *EvatCrop* increased significantly. The $Ag$ increased by 16.56%, and the $Ae$ dropped by 82.00%. The results presented in Tables S7 and S6 summarize that (i) *EvatCrop* is the most accurate model for estimating $ET_0$ in the testing set at Jayanti location; (ii) for combinations $C2$, $C6$, and $C8$, the DT model suffered from overfitting of data in the training phase, whereas the performance degraded in the testing phase; (iii) the ANFIS model failed to provide a good estimation for combinations $C7$ and $C8$ due to an out-of-range problem; and (iv) the input combinations with $W_s$ (combinations $C2$, $C5$, $C6$, and $C8$) yielded more accurate results than the other four combinations without $W_s$.

The scatterplots of the reference values of $ET_0$ ($ET_{0PM}$) and the predicted values of $ET_0$ are presented in Fig. S14 for the training and testing phases. Additionally, boxplots, Taylor diagrams. and distribution plots are provided for the testing phase in Figs. S15–S17, respectively.

## Estimating $ET_0$ of the Tamaguri location

From the results reported in Table S9, the *EvatCrop* proved to be the most accurate model during the training phase for all input combinations at the Tamaguri location. The ANN model was the least accurate model, whatever the input combinations. The ANFIS model displayed the same accuracy as that of *EvatCrop* for input combination $C8$ in terms of $Ag$. However, ANFIS was slightly inferior to *EvatCrop* for the combinations $C1$ to $C7$ in terms of $Ag$ (differences ranging from 1.50% to 4.00%) and $Ae$ (differences ranging from 3.31% to 3.96%), respectively. For combinations $C1$, $C3$, $C4$, and $C7$, the accuracies of the DT, ANFIS, and *EvatCrop* were found to be very low. In conclusion, the performance of DT, ANFIS, and *EvatCrop* varied marginally from one another in the training phase for almost all the input combinations.

The results obtained in the testing phase at the Tamaguri location are reported in Table S10. From Table S10, it is observed that, regardless of the input combinations, the *EvatCrop* was the most accurate model in terms of the highest values of $Ag$ and the lowest values of $Ae$. The performances of the ANFIS model were very poor against the combinations $C7$ and $C8$ due to the out-of-range problem ($Ag$ ranging from 0.548 to 0.948 and $Ae$ ranging from 0.110 to 0.558). But compared to the performance of ANFIS at Berubari and Jayanti, its performance was relatively acceptable ($Ag = 0.948$) at Tamaguri for combination $C8$. The main reason for this is the lesser number of outliers in the testing set, as observed from the boxplot (Fig. S19H). Contrary to the results obtained in the training phase, the ANN model performed comparatively better in the testing phase in terms of $Ag$ (difference ranging from 1.00% to 3.61%) and $Ae$ (difference ranging from 0.780% to 45.000%). The DT model ranked in third place, whereas the ANFIS was the worst.

To study the impact of the meteorological inputs on model accuracy at the Tamaguri location, the results obtained in the testing phase by using *EvatCrop* are summarised in Table S10. Using two inputs ($T_{min}$ and $T_{max}$) (combination $C1$), the *EvatCrop* exhibited its lowest accuracy ($Ag = 0.661$ and $Ae = 0.641$). When wind speed $W_s$ was considered with

$T_{min}$ and $T_{max}$ (combination $C2$), the performance of *EvatCrop* was remarkably improved. *Ag* was increased by around 47.66% from 0.661 to 0.976, whereas *Ae* was declined by 72.54% from 0.641 to 0.176. However, it is to be pointed out that the accuracy of *EvatCrop* decreased significantly when relative humidity ($R_h$) or solar radiation ($S_r$) was considered instead of $W_s$, with two inputs ($T_{min}$ and $T_{max}$) ($C3$ and $C4$). *Ag* values were reduced by 28.68% for combination $C3$ and 27.46% for combination $C4$. Accordingly, for combination $C3$, the *Ae* values increased by 70.96%, and for combination $C4$, the increase was 70.71%. In the case of four input combinations (combinations $C5$, $C6$, and $C7$), two combinations with $W_s$ ($C5$ and $C6$) yielded reasonably acceptable results. The *EvatCrop* scored an average *Ag* value of 0.984 against two combinations, $C5$ and $C6$, which was 41.34 and 38.98% better than combinations $C3$ and $C4$, respectively. Accordingly, it was found that the average value of *Ae* for *EvatCrop* against two combinations $C5$ and $C6$ was 0.141, indicating a significant decrease of 76.73% and 76.54%, respectively, for combinations $C5$ and $C6$. However, the accuracy of all the models was significantly decreased when $W_s$ was not included in the four-input combination ($C7$). The value of *Ag* was reduced by 23.82% when combination $C7$ was used instead of combination $C5$. Comparably, the value of *Ag* decreased by 24.75% when combination $C7$ was used in place of $C6$. From the results reported in Tables S9 and S10, it is concluded that (i) out of the four models, the *EvatCrop* turned out to be the most accurate model, (ii) the out-of-range problem prevents the ANFIS model from providing an acceptable accuracy for combinations $C7$, and (iii) input combinations with $W_s$ (combinations $C2$, $C5$, $C6$, and $C8$) yielded more accurate results than the other four combinations without $W_s$.

Figure S18 provides the scatterplots for estimated values against the reference values of $ET_0$ in the training and testing phases at the Tamaguri location. For a more intensive analysis of the results, the boxplots, Taylor diagrams, and distribution plots for the testing phase are also presented in Figs. S19–S21, respectively.

## DISCUSSION

In the present article, we have proposed a novel quasi-fuzzy ANN model, *EvatCrop*, for estimating $ET_0$ for the terai agro-climatic region of the North Bengal region, India. The estimated $ET_0$ values of *EvatCrop* have been compared with the reference $ET_{0PM}$ values under various input combinations. The $ET_0$ predictions of $ET_0$ are found to be highly acceptable for combinations $C2$, $C5$, $C6$, and $C8$, with an average $R^2$ value of 0.973.

Since $ET_0$ estimation is very region-specific, the *EvatCrop* was compared with the other relevant models reported in the literature for the Indian context. *Kumar (2023)* conducted an experiment with three machine learning models (Random Forest, gradient boosted trees, support vector machines) and one deep learning model (long-short term memory) for $ET_0$ estimation in data-scare conditions for Uttarakhand, India, and reported that the Random Forest and gradient boosted trees achieved the highest accuracy with an average $R^2$ value of 0.920 in the testing phase. Compared to this, *EvatCrop* recorded the highest average $R^2$ value of 0.982 for the test set. *Patle et al. (2023)* developed ANN and multiple linear regression models to estimate $ET_0$ in the semi-humid regions of Sikkim, India. The highest value of $R^2$ was reported at 0.820 in the testing dataset. *Ehteram et al. (2019)*

proposed an improved support vector machine model optimized using the cuckoo search algorithm for $ET_0$ estimation in Uttarakhand, India. The highest $R^2$ value reported was 0.944, which is less than that of *EvatCrop* (with $R^2$ = 0.982). *Patil & Deka (2017)* studied the performance of hybrid wavelet-ANN and wavelet-ANFIS for forecasting $ET_0$ in arid regions of India. The lowest value of *RMSE* reported was 0.586 mm/day for the wavelet-ANN model. In comparison, the lowest average value of *RMSE* was 0.201 mm/day. For the Northern Punjab region of India, *Saggi & Jain (2019)* reported a deep learning model for estimation of $ET_0$ that exhibited the best $R^2$ and *RMSE* values of 0.990 and 0.269 mm/day, respectively. Although the $R^2$ values of the deep learning model were better than *EvatCrop* ($R^2$ = 0.982), the performance of *EvatCrop* in terms of the *RMSE* metric was better with 0.201 mm/day. Some models had higher $R^2$ values than *EvatCrop*. These included the ANN-Grey Wolf optimizer hybrid model (with *RMSE* = 0.059 mm/day) suggested by *Tikhamarine et al. (2019)* and the ANN model created by *Heramb et al. (2023)* (with $R^2$ = 0.996). Overall, *EvatCrop* is a promising model with high accuracy for estimating $ET_0$ in the terai agro-climatic region of North Bengal, India, where no such models have been proposed to date.

# CONCLUSION

Precise estimation of $ET_0$ is a prime issue for efficient irrigation scheduling and better conservation of groundwater resources. In this article, we have suggested a novel architecture of a hybrid quasi-fuzzy ANN model for the estimation of daily $ET_0$ using prevalent metrological parameters. The efficiency and utility of *EvatCrop* were evaluated by using five prevalent meteorological parameters as inputs. *EvatCrop* was also compared with three other popular models, DT, ANN, and ANFIS, for eight different input combinations of five meteorological parameters at three study locations. By analysing the results obtained in the testing phase, the following conclusions would be drawn: the *EvatCrop* excelled over the other popular models, DT, ANN, and ANFIS, for the three study locations, irrespective of the input combinations of the meteorological parameters. For most input combinations at three study locations, the accuracy of ANFIS was slightly better than that of ANN, and the DT was the least performing model. Therefore, *EvatCrop* is the best model for the said region. In most cases, ANN ranked second, followed by ANFIS and DT. At the three study locations, the variation in the performance of all the models for estimating $ET_0$ is very similar, despite the variations in the input combinations. All the models provide their best accuracy when all five meteorological parameters are considered as inputs and perform worst when only two inputs ($T_{min}$ and $T_{max}$) are available. The accuracies of the four models are endurable for the three-input and four-input combinations only when the wind speed ($W_s$) is considered as one of the input parameters. This observation leads to the conclusion that the input combinations with $W_s$ yield more accurate results than the other four combinations without $W_s$, and it is the most contributing parameter for the estimation of $ET_0$ in this region. From the obtained results, we can further conclude that, due to an out-of-range problem, the ANFIS fails to provide good accuracy against a five-input and one of the three-input combinations at all three locations. This study provides a guideline for selecting the most promising input

combination of meteorological parameters to achieve the best possible estimation accuracy of $ET_0$ when a complete set of parameters is unavailable.

The limitations of this study are: (i) that our proposed model was trained and tested using datasets obtained from a particular region, and its performance in other agro-climatic regions was not evaluated; and (ii) that the impact of meteorological parameters other than the five meteorological parameters was not studied. However, our proposed model outperformed the other models in the Terai agro-climatic region of North Bengal, India. Further validation of our proposed model is required to ensure its superiority in other regions with meteorological parameters other than five prevalent meteorological parameters, which is our future target.

### Funding

The authors received no funding for this work.

### Competing Interests

Uditendu Sarkar is employed as Scientist-F at the National Informatics Centre, Ministry of Electronics & Information Technology, Government of India. Sanway Sarkar is employed as a Senior Consultant at Ernst & Young LLP, India.

### Author Contributions

- Gouravmoy Banerjee conceived and designed the experiments, performed the experiments, analyzed the data, prepared figures and/or tables, authored or reviewed drafts of the article, and approved the final draft.
- Uditendu Sarkar conceived and designed the experiments, performed the experiments, analyzed the data, prepared figures and/or tables, authored or reviewed drafts of the article, and approved the final draft.
- Sanway Sarkar conceived and designed the experiments, performed the experiments, analyzed the data, prepared figures and/or tables, authored or reviewed drafts of the article, and approved the final draft.
- Indrajit Ghosh conceived and designed the experiments, performed the experiments, analyzed the data, prepared figures and/or tables, authored or reviewed drafts of the article, and approved the final draft.

### Data Availability

The raw data are available in the Supplemental Files.

### Supplemental Information

Supplemental information for this article can be found online at http://dx.doi.org/10.7717/peerj.17437#supplemental-information.

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
