# Peer review of "EvatCrop*: a novel hybrid quasi-fuzzy artificial neural network (ANN) model for estimation of reference evapotranspiration"

_PeerJ, doi:10.7717/peerj.17437_

## Round 0.1 · original submission · Major Revisions

The introduction provides valuable background information on ET0 estimation methods. However, it could be shortened by removing redundant details. More focus should be placed on clearly introducing the gap/problem being addressed and the objectives/contributions of this study.

While the study area and datasets are adequately described, more details could be provided on the specific locations where data was collected and any preprocessing performed. Additionally, sample sizes for the training/test sets would help evaluate the model.

Lines 230-238 need citations like MLP’s background, previous used etc :

The explanation of the methodology could be enhanced by providing pseudocode for the quasi-fuzzy and ANN modules. Visuals like flowcharts would help readers follow the complex model architecture. How does the method compare with other machine learning?

The results section should report quantitative performance measures for each model on all input combinations/locations to better facilitate comparisons. Significance of findings also needs statistical tests.

The conclusions do not sufficiently discuss the limitations and assumptions of the study. Also, the practical/scientific impacts and potential applications of this model in ET0 estimation could be more strongly emphasized.

Some grammatical refinements are also needed. For example, redundant words and phrases like 'most noteworthy' and 'as depicted' could be removed for improved clarity and conciseness. Proper referencing format also needs verification.

Addressing these issues would help strengthen the technical soundness, rigor and communication of findings for the intended audience of this journal. The authors are encouraged to revise the manuscript based on these reviewer comments.

·

Basic reporting

The paper needs significant improvement

Experimental design

somme revision are necessary

Validity of the findings

the results are not well justified

Additional comments

Upon checking the manuscript, I can see that it suffers from serious weakness.

1. The authors have not justified the novelty of the paper
2. The writing style should be improved
3. In the introduction, literature review should be improved and recent published paper should be reported.
4. Novelty, research gap and the paper contribution needs to be clearly stated
5. Description of the data is unclear and needs more details
6. The section result is completely out of scope and cannot be accepted. The authors should presents the results in a clear manner. The results in the training and in the testing phase needs to be presented separately. I cannot understand what is reported in the Table, is the results for training or for the validation.
7. The quality of figure 4-6 needs improvement.
8. Boxplot, violinplot, Taylor diagram and necessary for improving the quality of the paper.
9. The paper is without section discussion. A separate section discussion should be provided and the comparison between the present paper and the results reported in the literature should be deeply presented and discussed.
10. Limitation of the study
11. Future recommendation’s.

Reviewer 2 ·

Basic reporting

In the study titled ‘EvatCrop: A novel hybrid quasi-fuzzy ANN model for estimation of reference evapotranspiration’ (#90695), the Authors estimated evapotranspiration (ET) using different methods and analyzed the performance. Studies on ET are valuable, and it is thought that the success and innovation of methods such as Fuzzy in this field make the study even more important. I think it would be appropriate to publish the article in the journal after making a few spelling corrections and additions to the models so that readers can understand it more clearly. I am giving minor revisions for the article and the corrections are given below:

Experimental design

1. Regarding spelling in the article, paragraph indents should be the same throughout the article. Some paragraph entries are placed 1 cm inwards, while others are aligned to the left.
2. In the introduction part, although the literature study of the article is well done, the fact that there are 10 different paragraphs makes the article appear unpleasant in terms of writing. Giving this section a maximum of 4 paragraphs, such as the introduction-body-conclusion, will make the article organized.
3. In the 'Datasets' section, it is mentioned that the data used are divided into training and testing, but the changes in these data are not given formally. In order to be well understood by the readers, the change over time of all data used (especially the estimated actual ET values) should be presented in the article.
4. Where the EvatCrop model is first explained, it should be explained why this abbreviation is given. Readers should understand why it was given this name.
5. Information about the methods is given under the headings ANN and Quasi-fuzzy, but what about the DP and ANFIS? It should be supported with a few sentences using a subheading.

Validity of the findings

6. While explaining the model, general and structural information is presented in the article. However, numerical parameters of the models used, limit values, step numbers, etc., other than MLPs. not mentioned in detail. For example, membership values for Fuzzy, shapes, fuzzy rules, etc etc. should be given in detail.
7. Scatter plots (Fig. 4-6) alone are not sufficient to compare model prediction values. However, distribution plots are also widely used in the literature. My advice to authors is to use distribution plots in future studies. Also, model inputs were repeated repeatedly in Fig. 4-6, and the model inputs were already given in Table 1. It should be given in abbreviation form according to Table 1. In all analysis figures and tables (Fig. 4-12 and Table 4-7), it is not stated that it is applied for the training or testing phase, whichever phase should be added.
For the information in 3,5-7, authors can review and benefit from the following academic studies:
……A comparative study on daily evapotranspiration estimation by using various artificial intelligence techniques and traditional regression calculations. Mathematical Biosciences and Engineering, 20(6), 11328-11352.
….. The evaluation and comparison of daily reference evapotranspiration with ANN and empirical methods. Natural and Engineering Sciences, 3(3), 54-64.
….. Evapotranspiration Calculation For South Carolina, Usa and Creation Different Anfis Models For ET Estimation. Aerul si Apa. Componente ale Mediului, 217-224.
8. Tables 6-7 are given separately. Just like Figures 4-6, Tables 5-7 should be given sequentially in the article.
9. When the metric results in the tables are examined, the units of the error, if any, should be added.
10. The model results, especially for the Anfis model, should be adequately discussed. In some combination cases, Anfis had the worst results, the reason should be examined. It should be examined whether the analysis was made using a grid partition or clustering partition for the lecture hall, and discussions can be made on similar issues.

---

## Round 0.2 · accepted · Accept

Thank you for your attention to the reviewer and editor comments provided on the original manuscript draft. The current draft has appropriately addressed those comments and I am recommending acceptance for publication.

Please consider one further modification for clarification of the paper structure: It seems that there may be a heading missing around line 445. There are headings for "Estimating ET0 of the Berubari location" and "Estimating ET0 of the Jayanti location" but no heading for the third (Tamaguri) location. For consistency of structure, that heading "Estimating ET0 of the Tamaguri location" should appear around line 445 and would improve the readability of the paper.

Reviewer 2 ·

Basic reporting

Since the authors have made the necessary corrections, it is appropriate to publish the article in its current form. I decide 'accept'.

Experimental design

Since the authors have made the necessary corrections, it is appropriate to publish the article in its current form. I decide 'accept'.

Validity of the findings

Since the authors have made the necessary corrections, it is appropriate to publish the article in its current form. I decide 'accept'.